

# A comprehensive review of security threats and solutions for the online social networks industry

Naeem A. Nawaz[1], Kashif Ishaq[2,3], Uzma Farooq[4], Amna Khalil[5], Saim Rasheed[6], Adnan Abid[7,8] and Fadhilah Rosdi[2]

[1] College of Computer and Information Systems, Umm Al-Qura University, Makkah Al-Mukarmah, Saudi Arabia
[2] Department of Informatics and Systems, University of Management and Technology, Lahore, Pakistan
[3] Faculty of Information Science and Technology, Universiti Kebangsaan Malaysia, Bangi, Malaysia
[4] University of Management and Technology, Lahore, Pakistan
[5] King Abdul Aziz University, Jeddah, Saudi Arabia
[6] Department of Information Technology, Faculty of Computing and Information Technology, King Abdulaziz University, Jeddah, Saudi Arabia
[7] Faculty of Computer Science and Information Technology, Virtual University of Pakistan, Pakistan
[8] Department of Computer Science, University of Management and Technology, Lahore, Pakistan

## ABSTRACT

The term "cyber threats" refers to the new category of hazards that have emerged with the rapid development and widespread use of computing technologies, as well as our growing reliance on them. This article presents an in-depth study of a variety of security and privacy threats directed at different types of users of social media sites. Furthermore, it focuses on different risks while sharing multimedia content across social networking platforms, and discusses relevant prevention measures and techniques. It also shares methods, tools, and mechanisms for safer usage of online social media platforms, which have been categorized based on their providers including commercial, open source, and academic solutions.

## INTRODUCTION

In this technologically advanced era, online social networks (OSN) have become quite accepted by the public. These enable individuals to socialize in various ways and naturally compel relations connectivity. It allows users to express themselves in virtual online communities to exchange information, share ideas, and interact with other users with similar interests. Despite that, several issues regarding managing user privacy and security may occur, particularly if multimedia content is uploaded. Online social networks connect to society's lifestyles as places of gatherings that encourage communication, which has caused the use of global OSN to grow exponentially. The social network establishes communication between the data owner and the viewer and creates a virtual community using the online communication network. A social network can be referred to as a social graph that shows the correspondence between users, organizations, and their social

Corresponding author
Adnan Abid, adnanabid7@gmail.com

activities. OSNs are online platforms end-users utilize to progressively construct social networks or connections of other people with similar perspectives, interests, activities, and contacts (*Shen, Ma & Eliassi-Rad, 2006*). A variety of social networking services are available in the present-day online space. The public is utilizing numerous OSNs for circulating multimedia. Because of this data usage, the majority are worried about the increasing security risks. It is known that malicious users can also produce harmful data and disguise it within multimedia data. Since they can get a hold of the user's identity and location, the attackers may try to take advantage, therefore, misuse it.

Even though a few OSNs do not permit a user to unveil crucial private information, the attackers can theorize the timeline of the posted known data of the user and leak the confidential information. To date, OSN, such as Facebook, Twitter, and others, were significantly attacked and had a notable consequence. Sometimes the attacker may not harm a general user's data, but they can still dangerously affect the OSN's general operations. Some services even transformed into the cause of illegal money-making. Following such events, it can be said that OSNs are the top means for malicious attackers to commit cybercrimes. Seeing this damage, the crowds bought the issue to the attention of some extent and security corporations and suggested several solutions. The solutions include steganalysis, watermarking, and data oblivion methods to protect the users against such harm. Many built-in solutions today, such as privacy settings and social protection applications, protect against these kinds of threats.

## Rationale of the study

Social media enhances the communication skills of students as well as provide the freedom to express in different ways like, posting a picture, audio, video, article, *etc*. It has several drawbacks including information theft by the hackers using new techniques such as social engineering, phishing, cat fishing, and many others. In this article, the authors discuses and combines the various types of threats and solutions in OSNs which is important for academicians, researchers, parents, students, and common person using social media must be aware of these security threats. These threats lead to loss of reputation, loss of intellectual property, data breach, compliance violation, and many others.

## Background

All existing OSNs accept client–server architecture and the OSN service provider operates as a controlling entity that manages all content in the program. On the other hand, users automatically generate this content, and the OSN service provider provides a rich set of well-defined areas that users can use to communicate with others. There are two known ways of communication. Facebook represents OSNs that embraces the interaction between senders and receivers as the main way they communicate, though they support alternatives. Twitter represents OSN that accepts broadcasting as its primary means of communication (*Rosenblum, 2007*). In 2005, YouTube launched, regenerating a completely fresh experience for people to interact over long distances. Upcoming, Facebook and Twitter were both available to users worldwide by 2006. These sites still have some of the most favored social networks to this day. Other sites like Tumblr, Spotify, Foursquare, and Pinterest are coming

out to fill specific social classes in societies (*Kaplan, 2010*). Currently, there are various social networking sites, many of which can be linked to grants for postage. This creates a certain environmental space where users can reach the maximum number of people without sacrificing the intimacy of the individual. We can only imagine what lies ahead for the prospects for social media over the next couple of decades.

### Social media on human communication

Social media has represented the most revolutionary social change in the history of communication, such as public relations, journalism, advertisement, marketing, entertainment, education, and businesses. Many ideas and concepts include dialogue, engagement, identity, social existence information, usage and satisfaction, a person's voice change, and many other social media. However, generally, what is happening in the context of social media and public relations is posted and replaces other ideas and concepts instead of exploring individual features and capabilities of social media. The document suggests that social media offers a new model for communication, and the text challenges the conceptualization of social media by identifying media as the basic research to understand current media to develop the communication theory (linguistics) of public relations (*GJJJ & Bello-Orgaz, 2016*).

### Social media and public communication

The notion of architecture is crucial to understanding events, decisions, and approaches, even in the social sciences, hard science, mankind, or any other field of study. Ideas, such as metaphors and self-help, but some differences suggest and refine existing ideas where there is the manifestation (*GJJJ & Bello-Orgaz, 2016*). Practical and nomothetic theories are presented in Table 1.

## Goals

Our goal is a survey of alternative threats that endanger privacy and security, directed at all users of social media sites. Furthermore, we focus separately on the risks of sharing multimedia content across a social networking site and other security techniques. We also study the behaviors of users of online social networks where users do not know how to access their information and integrate with surrounding information for multiple purposes. Efforts to mitigate this issue have focused on the individual aspects of consumer data collection. In this article, we also integrate tools to increase awareness of leaks of customer information, different threat solutions, and some techniques for protecting online social networks.

## RELATED WORK

Social networking sites have been growing over the last decade, and such forums have received much media attention. In *Al Hasib (2022)* and *Ishaq et al. (2022)*, the authors have revealed that users often disclose a variety of information, including their name, age, gender, address, photo, *etc.* some even hide and compile this information. As the use of social networks online is increasing in popularity, the security threats for users of these tablets have also increased dramatically. Many people find social networking to be

**Table 1  Practical and nomothetic theory.**

| Practical theory | Nomothetic theory |
| --- | --- |
| Axiological, Behavioral, or quantitative, Completed | Epistemological, about knowing and understanding the participants |
| Ontological, or based on current life, Feel social, volunteer, | Organization control- What you're watching, based on the question, |
| Prescriptive provides a guide Before the fact of how Behavior | Value is neutral, working for interests Empowering the team. It is not obligatory. |
| Cultural, focused Establishing relationships. | Organization/ Administration, Focus on reaching goals |

some of the techniques used for social networking online. Conditions have grown. People join Facebook to carry out three activities: 'like,' 'comments,' and 'shares.' Facebook uses that measure for different behavior to determine what is displayed on the user's screen, suggesting that the dynamic impact of each behavior may differ. This study investigates where each behavior can be promoted through organizational messages, and as a result, a clear difference has been made between these three behavioral factors (*Masrom et al., 2021*). Table 2 presents the comparison of this survey with other related works on online social networks.

# METHODOLOGY

This survey implemented recommendations for systematic reviews given in information engineering analysis by *Brereton et al. (2007)*. Based on these criteria, a search method was defined to eliminate possible study biases after finalizing research queries. Within this procedure, three critical phases of our analysis approach were to prepare, conduct, and review the study, as shown in Fig. 1 and discussed in the following sections.

## Review plan

For all relevant studies, a suitable search technique was developed. Figure 1 depicts how the analytical techniques describe the categorization scheme and item mapping, indicating the search operations for the related articles. This study adopted an organized approach (*Ishaq et al., 2021*):

- Research objectives
- Specifying research questions (RQs)
- Organizing searches of databases
- Studies selection
- Screening relevant studies
- Data extraction
- Results synthesizing
- Finalizing the review report

Nawaz et al. (2022), *PeerJ Comput. Sci.*, DOI 10.7717/peerj-cs.1143

**Table 2  Related work.**

| Ref. | Focus on Survey | Perspective Addressed OSN | | | | | |
|---|---|---|---|---|---|---|---|
| | | Newest Ref. | Details of Threats on ONS | Describe Techniques of Online Social Network | Privacy Protection in OSNs | User Behavior in OSNs | Threats solution |
| *Aleroud & Zhou (2017)* | Phishing environments, techniques, and countermeasures | 2017 | ✓ | ✓ | X | X | ✓ |
| *Rathore et al. (2017)* | Social network security: Issues, challenges, threats, and solutions. | 2014 | ✓ | ✓ | X | X | ✓ |
| *Adewole et al. (2017)* | Age Differences in Online Social Networking: Extending Socioemotional Selectivity Theory to Social Network Sites. Journal of Broadcasting & Electronic Media, | 2015 | X | X | ✓ | ✓ | X |
| This Survey | Security and Privacy issues in online social network | 2022 | ✓ | ✓ | ✓ | ✓ | ✓ |

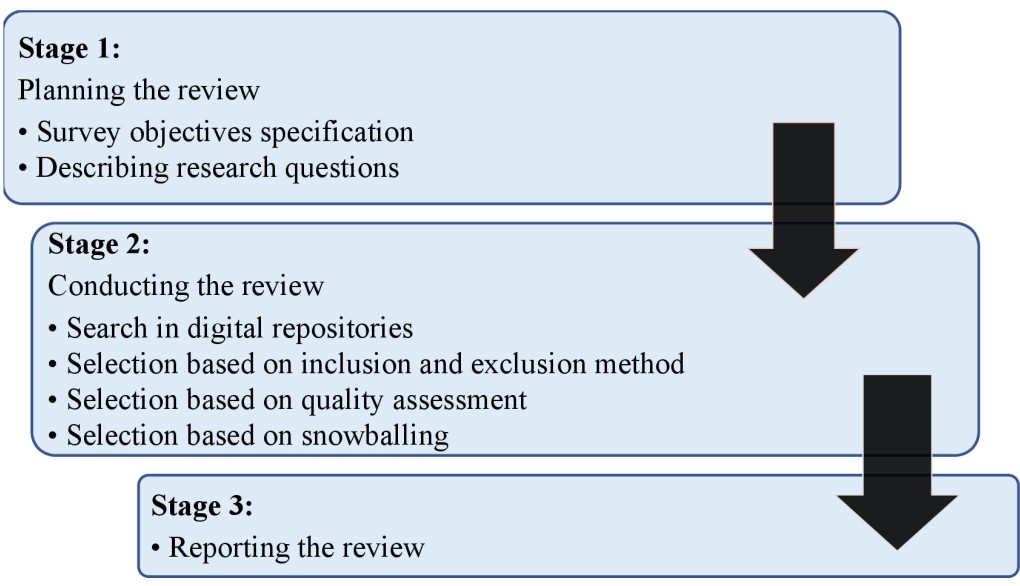

**Figure 1** Research strategy.

**Table 3** Research questions.

| Research questions | Motivations |
|---|---|
| Research Question 1:<br>What security threats are there on OSN which affect the OSN's users? | ● Classical Threats<br>● Modern Threats<br>● Combination Threats |
| Research Questions 2:<br>What techniques and solutions are used to secure online social networks, and what are their limitations? | ● Illegitimate user detection<br>● Authentication mechanisms<br>● Privacy and security settings<br>● Cluster Techniques<br>● Anomaly Detection Techniques |
| Research Question 3:<br>How to evoke awareness in users about privacy protection in online social networks? | ● Privacy behavior of OSN users<br>● Visibility of user's profile<br>● Privacy policy of OSN |

## Defining question

Table 3 illustrates the appropriate research questions (RQ) that have been developed from the study aims and objectives.

## Search strategy

The articles are relevant to the scope of our article, and the research articles are collected from online digital repositories such as ACM Digital Library, IEEE Xplore, ScienceDirect, and Google Scholar because these repositories are considered highly ranked and authentic in the researcher's community. The keywords used for searching articles are 'Different Behaviors of Users in Online Social Network,' 'Multiple threats to an online social network,' and 'Techniques used to protect the online social networks.' After searching the relevant

| Table 4 Search engine. | |
| --- | --- |
| **Search engine** | **Search content** |
| Google Scholar | • Multiple threats of an online social network. |
| | • Several techniques are used to protect online social networks. |
| Science Direct | • Multiple threats of an online social network. |
| | • Several techniques are used for protecting online social networks. |
| IEEE Xplore | • Different Behaviors of Users in Online Social Network. |
| | • Multiple threats of an online social network. |
| | • Several techniques are used to protect online social networks. |
| ACM Digital Library | • Different Behaviors of Users in Online Social Network. |
| | • Multiple threats of an online social network. |
| | • Several techniques are used to protect online social networks. |

articles, there was further analysis to issue our relevant articles. Table 4 represents the search engine and search content.

## Selection based on inclusion/exclusion criteria

The article comprised in the review must be in OSN that must target the research questions and the article published in the journals or conferences also from 2016 to 2021 was included in the review. Articles discussing ONS threats and techniques to protect the online social network environment included in the review. The articles were excluded not written in English and did not discuss or focused OSN and the techniques. A selection process of relevant articles for inclusion/exclusion criteria in detail was shown in Fig. 2.

## Selection based on snowballing

Following the standard assessment, backward snowballing was performed, employing a reference list from any completed analysis to retrieve articles (*Mehmood et al., 2020*; *Ishaq et al., 2021*) and chose only those significant articles that met inclusion/exclusion requirements. After reading the introduction and then other portions of the document, the article's inclusion/exclusion was determined.

### Overview of intermediate selection process outcome

OSN is a very active topic, and our analysis approach had to extract relevant research empirically and systematically from various databases like: the Web of Science, Google Scholar, IEEE Xplore, *etc.* The next step of our systematic analysis was compiling records that form the foundation for this analysis. More than 57,000 articles were examined from the archive by providing the keywords for 2016–2021. After creating a knowledge base from the digital libraries, the author reviewed the title, abstract, and accompanying complete document for each search result, as needed. During this process, irrelevant articles or articles of less than four pages were eliminated. During the inspection process, selected documents in the fields of OSN and its security techniques were read extensively to assess their significance & contribution and then created a comprehensive knowledge foundation of articles based on their findings to accomplish this research's core objective (*Ishaq et al., 2021*).

- **Identification**
  - Record Identified through databases search (n=55,461)
  - SCI-Expended; SCIE; ESCI; A&HCI; Google Scholar, IEEE Xplore
  - Record excluded for out of scope (n=51,637)

- **Screening**
  - Record screened by title (n=3824)
  - Record excluded (n=3622)
    - Out of scope title
    - Did not defined OSN

- **Eligibility**
  - Record Screened based on Introduction and Conclusion (n=202)
  - Record excluded (n=159)
    - Focus is not discussing OSN

- **Synthesis**
  - Studies included in the systematic review (n=43)

**Figure 2   Inclusion and exclusion.**

**Table 5   Selection phases and results.**

| Phase | Selection | Selection Criteria | Indexes: SCI-EXPANDED, SSCI, A&HCI, ESCI, Google Scholar, IEEE Xplore etc. |
|---|---|---|---|
| 1 | Search | Keywords (Figure) | 57,364 |
| 2 | Filtering | Title | 3,897 |
| 3 | Filtering | Abstract | 275 |
| 4 | Filtering | Introduction and Conclusion | 171 |
| 5 | Inspection | Full Article | 43 |

### *Overview of selected studies*

Table 5 presented significant results of the primary search, filtering, and reviewed processes that included digital libraries indices (*Ishaq et al., 2021*). At the filtering/inspection stage, this amount decreased to 43 articles by the automatic search.

# WHAT SECURITY THREATS ARE THERE ON OSN WHICH AFFECT THE OSN'S USERS?

OSN users are increasing daily and showing all their personal life, like family photos, location, and experiences, on such platforms. The online social network makes people connect to their friends and family. They share their information or activities through social media and publicize them when they connect. In *Clement (2020a)*, it is reported that total users of social media like Facebook, YouTube, Instagram, WhatsApp, Twitter, IMO, and Skype. The main issue of OSN users is security. Many components of social networks are given in *Rathore et al. (2017)*, such as contacts, social media, virtual community, relationships, connections, friendships, being online, and the Internet. Many threats involve using social media, creating security problems for the users. There are five categories of threats: Classical threats, Modern threats, Insider threats, Multimedia threats, and Targeting children. In *Clement (2020b)*, it is shown how the total number of users increases each year until 2020. The block diagram of all threats of online social networks and their categories is represented in Fig. 3.

## Classical threats

Since the establishment of the Internet, classical threats have long been a worry. These threats are the main problem for OSN users. Through classical threats, hackers or attackers collect personal information and misuse it. Many types of attacks are involved in it, whereas attackers can access other important information, like information based on a bank account by scanning personal information, and can commit online crimes, such as bank fraud. ONS attacks range from account stealing, fraud, and phishing attacks to malware distribution. Different types of classical threats are given below that imperil the privacy and security of users.

### Malware

Malware or malicious software is a computer program that harms the computer without users' knowledge, involving worms, viruses, Trojan horses, *etc*. Intruder uses malware intentionally to damage the computer or use the data for its purpose. Malicious software slows down the entire system and infects the operations. In online social networks, the malware uses the networks and users' information by sending messages to their friends to obtain their credentials. Attackers make malicious accounts to get their purpose because malware attacks are easier in online social networks than in other networks due to the structure of OSN (*Cao et al., 2014*). If malicious software has one or more functions, the threat level will increase, such as P2P-Worm, IRC-Worm IM-Worm, *etc*. Internet and e-mail are the two ways the virus originates, whereas malware constantly evolves and becomes more dangerous over time.

### Phishing attacks

Phishing is a fraudulent attempt to disguise itself as a trusted electronic communication center and obtain sensitive information such as usernames, passwords, and credit card details. The attacker sends the forged e-mail to the user as a weapon, like an e-mail about the

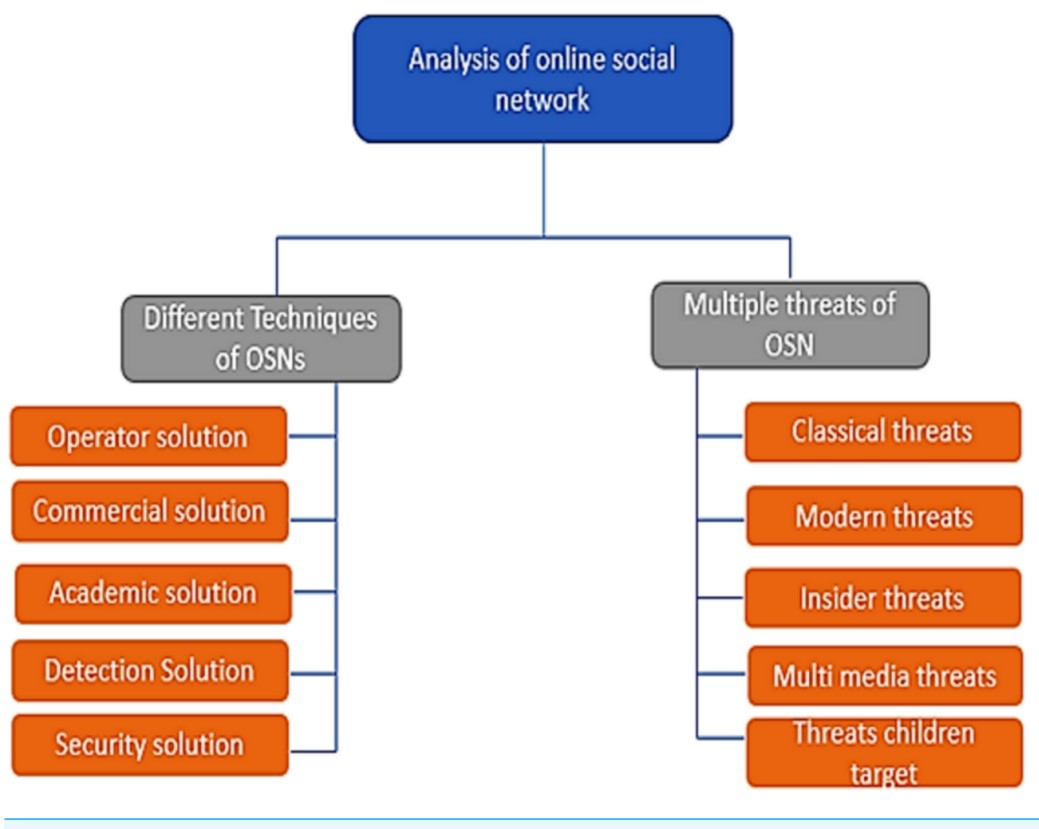

**Figure 3** Types of threats and solutions.

bank account, a message from a friend, or a note from the company. The attacker's purpose is to obtain the user's sensitive information. An attacker can send the link to the login page of the legitimate website where the user puts the name and password, then the attacker easily takes that information and use it improperly. In *Brad (2019)*, a simple animated phishing attack is explained with the help of a diagram that shows that the phishing attack process consists of five phases, *i.e.,* attack planning, setup, execution, fraud, and post-attack phases. These phases can be divided into sub-processes (*Aleroud & Zhou, 2017*): Attack Preparation, Attack Execution, and Attack Results Exploitation. There are three categories of target environment of a phishing attack: Smart devices and voice devices such as a desk or softphones and personal computers (PC). Attackers have many techniques to attack the users, and there are three categories of attack techniques: Initialization, Data collection, and System penetration.

### Spam attack

Spam is electronic junk mail or junk newsgroup posts. The attacker sends the unsolicited messages (*Cao et al., 2014*) (such as e-mails, text messages, or Internet postings) in bulk, often using a stolen mailing list that includes the user's address. The contents of these e-mails are complete gibberish. Spam attackers shift the cost of advertising to the victims, and spam messages are like unwanted messages or calls to your phone. The attacker's

purpose is to make money or profit from the users. Massive mailing is cheap and easy compared to physical mail. Every day 293.6 billion e-mails are sent, out of which 54 billion are spam messages. Social media attracts many spammers and is an ideal platform for spammers to attack or send inappropriate content and messages (*Fu et al., 2018*). Malicious spammers send irrelevant content on social media like commercials or links. Not everyone becomes a victim; spammers are the most inventive in creating the 'bait' for the users. When users get caught in that trap, the scammer gets all the information about that victim. There is some damage caused by spamming, such as irritation and discontent, waste of time, criminalization of spam, communications overload, and the loss of important e-mails.

### Cross-site scripting (XSS)

Cross-Site is a type of injection in which malicious scripts are disguised on the websites. XSS is a URL-based attack (*Yassein, Aljawarneh & Wahsheh, 2019*). XSS attacker sends the malicious code to the web pages, which is delivered with some content to the user's browser. The code which is sent to the user is in the form of bits of JavaScript code. In XSS, the attacker executes the malicious script in the user's browser, but the XSS attacker does not attack directly. The attacker sends the code to the website that the user visits and exploits the vulnerability in the website. It is the benefit for the attacker that the website does not tell whether the markup text is valid or not; it just executes it. The attacker injects the script into the webpage and uses it for its purpose, like the user's login page information, stealing the session cookies, *etc*. A three-step explanation of a cross-site scripting attack consists of 'attacker sends the malicious link,' 'User clicks the link,' and 'browser sends the private data to the attacker'; hence the attackers steal the user's sensitive information by sending the malicious link.

## Modern threats

Modern threats are associated with OSNs. It involves many threats that get the sensitive information of the user. For example, the attacker sends a friend request to a user on Facebook, and if the user's information is public, then the attacker sends the request to their friends, and if they accept, all the information is disclosed in front of the attacker. Facebook users post their location or any activity that shows they are not at home, and attackers can use all that information in the wrong way. Modern threats involve many threats, which are given below:

### Clickjacking

Clickjacking is a technique of tricking the user for which an attacker uses transparent layers. In *Banach (2019)*, an illustration of a transparent layer is shown with the help of a diagram that shows account-based and URL-based threats. When the user clicks on a link, that takes the user to the attacker's destination route. Users may not know that click takes to any other route. The attacker manipulates the user into posting spam messages on their Facebook or Twitter timeline (*Fire, Goldschmidt & Elovici, 2014*). The common clickjacking involves a login and password page. The user thinks that id or passwords are entered in the legitimate website, but that website is just the layer on a user interface (UI),

and a user enters the information in the hacker's fields. Hacker/attacker uses that password or credit card number for their benefit because all that is about money, property, revenge, or fun.

### De-anonymization attacks

De-anonymization is a technique in which an attacker tries to assume the user's identity from traces (*Gambs, Killijian & del Prado Cortez, 2014*). Information is being re-identified in this technique. Re-identification is the procedure by which anonymized individual information is matched with its owner. In numerous OSNs such as Facebook and Twitter, users choose to shield and secure their identity through various privacy settings. Users make their social accounts with alias or false names to protect their privacy. Attackers use the cookies to uncover the user's real identity, whereas other methods are user group membership and network topology. OSNs are an easy target for de-anonymization attacks because most data are shared through social media. The attacker can link the information which the user on OSN reveals.

### Sybil attack or fake profile

In Sybil or face profile, the attacker makes multiple fake identities of the user, damaging the reputation system's power. Fake profiles are used to initiate the Sybil attack. It is named after the subject of the book Sybil, a case study of a lady determined to have a dissociative personality issue. One can have many fake followers and tweets in a fake profile; also, the number of likings can be increased and boost the artificial popularity on any social media. The attacker has many fake accounts to perform malicious activities. In this attack, an attacker sends messages to legitimate users, collects their personal information (*Ali et al., 2018*) for only friends and family, and spreads that information as spam. But that fake profile has one or no posts on their timeline. If the user has ten or more than ten friends, one of them accepts the attacker's request, then all other friends will be affected. This attack can reduce the value or reputation of users and spread false information.

### Identity clone attacks

OSNs' main problem is authentication. People want to connect with their friends and family members, but they do not know whether the account or id is authenticated or not. In this attack, the attacker replicates the user identity to make a relationship with friends in the same network or another network (*Malenkovich, 2012*). The attacker makes the fake accounts, takes all the user information (*Kharaji, Rizi & Khayyambashi, 2014*), and uses the trust of the cloned user to exploit. Identity clones have been the source of much fraud. This attack affects the relationship of the user with others. On OSNs like Facebook, Instagram, Twitter, *etc.*, the attacker makes the several identities of many celebrities to take many benefits like increasing their followers, likes, comments, or getting the trust of other users.

### Inference attacks

An inference attack is a data mining technique in which an attacker illegitimately takes the user's information. Everyone except their family and friends hides the information of

the user. The attacker used the user's friends' attributes to predict the user's attributes. The hidden information is sensitive and cannot be revealed by everyone, and the attacker used data mining techniques to disclose all the sensitive information. OSN is the biggest platform for the attacker to attack several users because each user has a friend list (other victims) and behavior digital record, behavior record including the pages they like or the applications they installed on their mobiles (*Gong & Liu, 2018*). The attacker took the benefit of all the information and used it for his purpose.

### Information leakage

On social media, users can openly share their information with friends and family or make that post public, where everyone can see and download it. Sometimes, users share sensitive information with their friends like passwords of accounts, health issues, information about family, *etc*. Information can still be leaked if the user sets a privacy setting (*Li, Yan & Deng, 2015*). Many users share too much sensitive information on OSN, which negatively impacts insurance companies, considering them risky clients. People who want a job and send their resume to an organization can check their OSNs accounts and find their bad habits.

### Location leakage

Mobile devices make it easier for the attacker to locate any person. With the advancement of technology, more users use mobile devices to connect to social media and reveal their location. For example, users send photos in restaurants or places with geotagging information that shows their location for which attackers can take its benefit to harm them. There are two types of location attackers: Causal Localization and Determined Tracking attackers (*Li et al., 2014*).

### Surveillance

Surveillance is like monitoring someone's activities, information, or behavior for some purpose. Whatsoever the purpose, surveillance itself is inferior. On OSNs, an attacker can monitor the user's posts, likes, and followings to extract the information. WhatsApp discovers that the attacker calls the user; if the user does not pick up the call, surveillance software will be installed on the user's device (*Lee, 2019*).

## Insider threat

An insider threat is a malicious threat in which the attacker is a known individual to the user or any organization. On OSN attacker can be a friend or any person close to the user, know about the id and password of the user, and login into the victim's account in an authorized way. When an attacker gets the authorization of the victim's account, they use all information like the victim's private information, a friend list, or information about their friends. They then use that account to damage the victim's reputation (*Usmani et al., 2017*). Insider attackers can damage the whole organization as well (*Homoliak et al., 2019*). There are five examples of insider threats (*Chavali, 2018*) that caused breaches in organizations in Table 6.

Anthem's huge data breach was hit with an insider theft, which resulted in the stealing of personal data for more than 18,000 members of Medicare. In April 2017, Anthem's

| Table 6 Examples of insider threats. | |
|---|---|
| **Anthem** | **Employee data ex-filtration** |
| Target | Third-Party Credential Theft |
| RSA | Employees Fall for Phishing Attacks |
| Sage | Unauthorized Employee Access |
| Boeing | The Nation-State Spy |

Medicare insurance coordinating services vendor heard of a worker who thieved and had been held accountable for misusing Medicaid patient data since July 2016. Among other infractions, the guilty employee had sent to his e-mail address a file containing data about Anthem members. The data included Medicare ID numbers, Social Security numbers, Health Plan ID numbers, members' names, and registration dates. The worker was fired for this and other matters, which were therefore investigated.

Target's widely publicized 2013 breach of credit card data resulted from a third-party vendor that is some other form of insider threat, getting a hold of sensitive system credentials out of a suitable use case. The hackers are granted to take advantage of Target's payment systems vulnerabilities through credential access to acquire access to a customer database and install malware. They were then capable of stealing Target's customers' personal identifiable information (PII), involving: names, phone numbers, e-mail addresses, specifics of the payment card, credit card authentication codes, and more. Without a doubt, the violation was unfortunate news for Target, but it acted as a warning of caution to other companies.

Two hacker groups affiliated with a foreign government conducted phishing attacks on employees at RSA in March 2011, claiming to be trusted colleagues and contacts. The hackers gained access when the workers fell for the attacks and were able to compromise SecureID authentication tokens. The hackers gained access when the workers fell for the attacks and were able to compromise SecureID authentication tokens. One of the most surprising aspects of the attack was that RSA as a security vendor has long been highly regarded. The attack revealed that none are resistant to data breaches caused by insiders.

Sage, a UK-based accounting and HR software provider, was hit with an insider-caused data breach in 2016, which compromised 280 of its company clients. A woman who worked for the organization used unauthorized access to steal information about private customers, including salary and bank account information. Although the violation was fairly small, it highlights the issue of insiders who can gain access to highly sensitive customer data–allowed or not.

Greg Chung, who spied for China while working at Rockwell and then at Boeing, stole from 1979 to 2006, when he was eventually arrested, stealing hundreds of boxes full of military and spacecraft articles. There might be no way of putting a dollar figure on the uncountable amount of stolen data or the implications of its theft.

## Multimedia threats

A large amount of data is being shared on OSN, specifically multimedia content like videos and images. OSN allows the users to share multimedia content in high resolution, but

that high quality can bring some threats like location information through geotagging, face recognition, and home address; all that information revealed may harm the users. For example, if a user shares their photos publicly, an attacker can use that photo for an identity clone and use it for a biometric database revealing the home's location for any damage. Many people publicly post their photos on social media on Facebook, Twitter, *etc.*, to shape their memories (*Dhir et al., 2016*, *Bartsch & Dienlin, 2016*). Users post their photos with their friends without any privacy settings that can be used in a biometric database and identify the social media use without permission. It becomes a biometric threat for the users. Attackers or hackers can use the user's photo to steal their identity for different purposes like obtaining their security number, activities, *etc.*, whereas three experiments have been done on threats of face recognition (*Fire, Goldschmidt & Elovici, 2014*). There are many types of multimedia threats: multimedia content exposure, shared ownership, steganography, metadata, static links, outsourcing and transparency of data centers, video conference, Tagging- linkability from shared multimedia data, and unauthorized data disclosure (*Rathore et al., 2017*).

## Threats targeting children

Children or teenagers use the Internet more actively and are mainly subjected to threats presented in *Fire, Goldschmidt & Elovici (2014)*. Online Predators, Risky Behaviors, and Cyberbullying are examples of children-related threats. There are some other threats targeting children or teenagers given in the below section.

### *Harmful or aggressive content*

OSN is the platform of free expressions and opinions that brings the darker side of the web (*Keipi et al., 2017*). Everyone has the right to speak openly, but sometimes people speak aggressively or share content that is not appropriate that could be more harmful, especially for children or teenagers. Children or teenagers pick everything more quickly without thinking or unconsciously.

### *Scams*

Recently scams have been growing day by day. Scammers contact children through many sources like text messages, e-mail, and social media, the biggest platform for children. Scammers make the commercial more attractive to children or teenagers because they are vulnerable to scammers. Scams may be the winning scams; send messages to them or make commercials publicly, so children or teenagers could see or take an interest in them (*TheStreet, 2010*).

### *Fake friends entering the chat room*

Attackers make fake accounts, pretending to be their age to enter the chat room. Firstly, the attacker tries to be friendly, giving them full attention and pretending to be concerned about them. When attackers have more information to threaten them, they reveal their true identity and threaten them with information or anything they got from them (*Norton, 0000*).

# WHAT TECHNIQUES AND SOLUTIONS ARE USED TO SECURE ONLINE SOCIAL NETWORKS, AND WHAT ARE THEIR LIMITATIONS?

The expansion of online social media networks has increased privacy and security issues. Therefore, preventive measurement has been done to overcome these issues and threats to OSN. There are many proposed techniques, solutions as well as models that lower the possibility of threats. The solutions that help to eliminate the threats are listed in Table 7. In this section, we describe the solutions for OSN threats.

## Operator solutions

By doing protection measurements, OSN users can protect themselves from threats. The main diagram of operator solutions and their categories can be visualized in Fig. 4. The description of operator solutions is presented in Table 8.

### Authentication mechanism

OSN and other websites use authentication methods, so unauthorized users cannot access the social network. Facebook, Instagram, Twitter, *etc.*, all these social media accounts need login and password to protect the user account. Moreover, these platforms adopt other authentication mechanisms like sending verification codes and Captcha for privacy (*Fire, Goldschmidt & Elovici, 2014*). A verification code sends to the user by e-mail or on a registered mobile device. Mostly these verification codes are integer numbers; for example, when the user creates a WhatsApp account, the verification code is sent to the user's mobile device to complete the registration process. Similarly, many websites have Captcha, in which random pictures must be used to complete the verification process. In Gmail, entering the wrong password several times alerts the secondary user of the e-mail with the wrong attempts, including the city name, by the Gmail operator immediately. There are three batch authentication protocols.

(a) *Hash-based authentication protocol:* A hash-based authentication protocol is suited for resource-limited devices and needs less computational cost.

(b) *Proxy-based protocol:* This protocol is used for exchanging information between users and is based on asymmetric encryption.

(c) *Certificate-based protocol:* Certificate-Based protocol assures non-repudiation of transactions by signatures (*Facebook Immune System , FIS*).

### Security and privacy setting

Security and privacy settings are included in all social media networks. Users can set the privacy setting in their account to protect it (*Fire, Goldschmidt & Elovici, 2014*). For example, on Instagram, users can do some security measurements like one can hide their story and post to other users for safety, or users can block them. On Facebook, users can do some privacy settings like who can see the status, profile photo, likes or dislikes, friend list, *etc*. With these privacy settings, user can protect their accounts from attackers.

**Table 7  Description of threats and solutions.**

| Types | Description |
|---|---|
| **Operator Solutions** ||
| Authentication Mechanism | Verification code, Captcha for privacy terms, login, and password are used to authenticate the user. |
| Security and Privacy setting | Users can set the privacy setting in their account, like who can see the post, block users, etc. |
| Internal Protection Mechanism | Protecting users from spammers, identifying fake profiles, and scammers. |
| Report Users | Users can report another user on social networks like Facebook, WhatsApp, Instagram, etc. |
| **Security Solutions** ||
| Watermarking | Watermarking is a form of embedded data into media content to show media content's ownership. Including watermarks in the multimedia file enables a user to track certain activities as if other users were re-uploading or updating their multimedia file in OSNs. |
| Co-ownership | In OSN, Co-ownership allows various users to carry out the privacy policies on the co-owned videos and pictures. |
| Steganalysis | Steganalysis is a method to detect harmful data within multimedia. |
| Digital Oblivion | On digital data, the expiration time is placed because other users are forbidden to access data when the expiration time of data is over. |
| Storage Encryption | Storage encryption allows the OSN users to recover and store the data without exposing the data to others. |
| Metadata Removal and Security | The solution provides various approaches to the elimination of metadata and reducing the privacy leakage in OSNs |
| Intrusion Detection | An intrusion detection system is used to analyze the actions of users and the system. Principal Component Analysis (PCA) analyzes user behavior and detects anomalies |
| **Commercial Solutions** ||
| Internet Security Solution | Companies like McAfee, Panda, Symantec, etc., offer antivirus and firewalls that protect the user's system. |
| AVG Privacy Fix | Facebook, Google, and LinkedIn used AVG privacy Fix software to protect users' privacy. |
| FB Phishing Protector | FB Phishing Protector is a tool of Firefox. This tool prevents the users on Facebook from Phishing scams. |
| Norton Safe Web | Norton Safe Web is used to identify websites with malicious content or links. |
| McAfee Social Protection | McAfee Social Protection allows users to share photos only to selected friends on Facebook. |
| MyPermission | MyPermission is a privacy cleaner and also discovers threats to users' data. |

**Table 7** (*continued*)

| Types | Description |
|---|---|
| **Operator Solutions** | |
| NoScript Security Suite | No Script Security Suite has a blocking approach that blocks the malicious scripts running on the browsers. |
| Privacy Scanner for Facebook | Reclaim Privacy and Trend Micro Smart Protection scanner are used for Facebook to check the privacy setting and block malicious activity and links on Facebook |
| Defensio | Defensio is a web security service that removes malicious content, spam, and comments from the web. |
| ZoneAlarm Privacy Scan | ZoneAlarm is Anti-Phishing Privacy Scan that is used to detect phishing attacks. |
| Net Nanny | Net Nanny is software used to block inappropriate content. |
| Minor Monitor | Minor Monitor is a service used by parents to track their children's Facebook activities and monitor them. |
| **Academic Solutions** | |
| Improving Privacy Setting Interfaces | Facebook offers the function in which one user sees the interface of their account that an unknown user can see. |
| Phishing Detection | Countermeasures for phishing attacks include Machine Learning, Text Mining, Human Users, and Profile Matching, algorithms like Random Forest, Naïve Bayes, Adaboost, Decision Tree, SMO, and K-star. |
| Spammer Detection | A community-based feature for identifying spammers and a Particle swarm optimization algorithm are used to detect spam on Facebook. |
| Cloned Profile Detection | Two methods are used Basic, i.e., Profile similarity, and multiple fake identities profile similarity |
| Sybil Detection | Random Forests, Naive Bayes, Adaboost, and SVM, detect Sybil attacks. |
| Detection of information and Location Leakage | BMobishare is used to preserve the information of location. Twitter offers the guardian angel service that tells the user if any privacy violation occurs. |
| **Malicious Account Detection Solutions** | |
| Crowdsourcing | Crowdsourcing involves crowd workers checking users' profiles to detect malicious activity. |
| Graph-Based | Graph identifies the malicious behavior and malicious accounts on OSN |
| Supervised Learning | Bayes theorem, Meta Based, Support Vector Machine, Neural Networks, and Tree-Based methods are used to detect malicious accounts. |
| Unsupervised Learning | Unsupervised learning uses an artificial intelligence algorithm to detect the variations in the dataset for malicious accounts and user behavior. This method uses hierarchical, Partitional, PCA-based, Stream-based, and Pairwise Similarity methods. |
| Semi-supervised Learning | Semi-supervised learning uses the TSVM algorithm is used to detect the phishing attacks |

**Table 7** (*continued*)

| Types | Description |
|-------|-------------|
| **Operator Solutions** | |
| **HRSP Model** | |
| HRSP URL Classification Model | URL lexical features are used to detect URL-based threats like phishing and XSS. Blacklist-based solutions also detect malicious URLs. |
| HRSP Content Classification Model | The content classification model is used to analyze the user's content, like posts and comments, to eliminate the threats. HRSP content classification model uses the machine learning techniques like Offensive Dictionary, Users' Feedback, and Inappropriate Dictionary. |

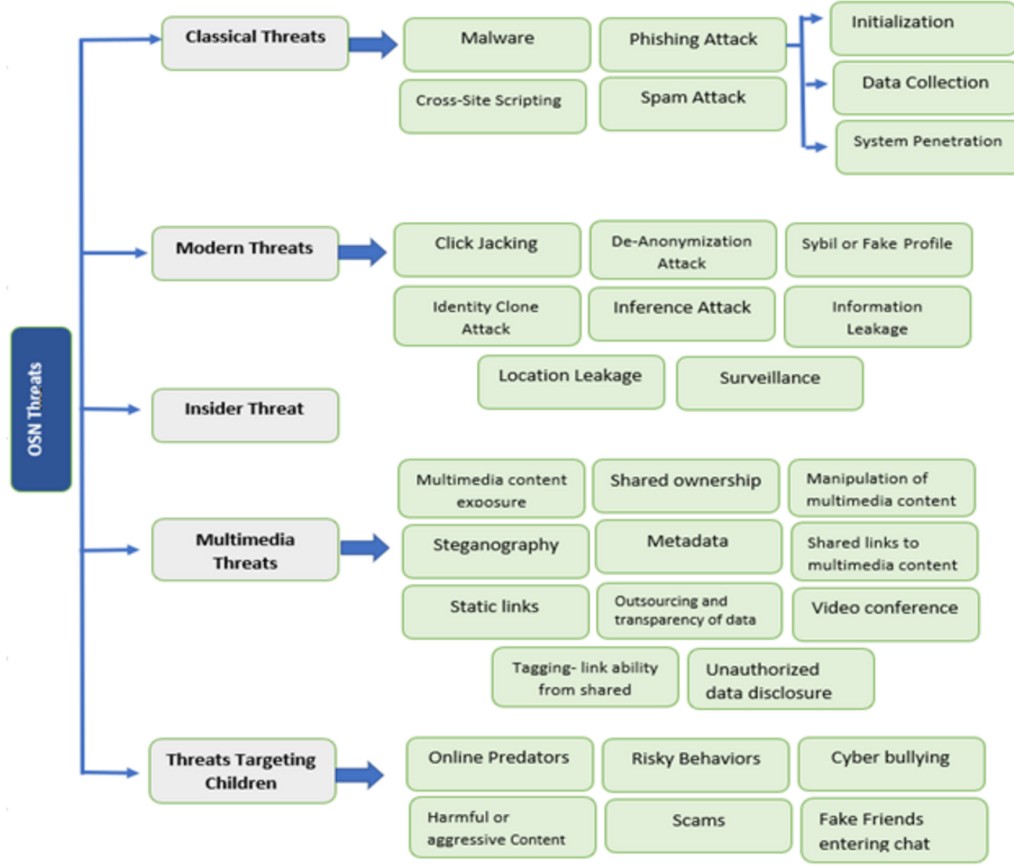

**Figure 4** Block diagram for OSN threats.

## Report users

Users can report another user for many reasons, whereas online social media accounts give the facility to the users of reporting. These users can utilize that reporting function for their interests on Facebook, WhatsApp, and Instagram.

**Table 8 Solution of threats.**

| Threats / Solutions | Spammers | Phishing Attacks | Clickjacking | Internet Fraud | De-anonymization | Malware | Cross-Site Scripting Attacks XSS | Identity Clone | Sybil or Fake Attack | Information Leakage | Location Leakage | Inference attack | Online Predators | Risky Behaviors | Cyberbullying | Multimedia content exposure | Shared ownership | Manipulation of multimedia content | Steganography | Metadata | Shared links to multimedia content | Static links | Outsourcing and transparency of data | Video conference | Tagging-linkability from shared multimedia data | Unauthorized data disclosure |
|---|---|---|---|---|---|---|---|---|---|---|---|---|---|---|---|---|---|---|---|---|---|---|---|---|---|---|
| Phishing Detection | ✓ | ✓ | ✓ | ✓ | | ✓ | | | ✓ | | | | | | | | | | | | ✓ | | | | | |
| Spammer Detection | ✓ | ✓ | ✓ | ✓ | | | | | ✓ | | | | | | | | | | ✓ | | ✓ | | | | ✓ | |
| FbPhishing Protector | | ✓ | ✓ | | | | ✓ | | | | | | | | | | | | ✓ | | ✓ | | | | | |
| Cloned Profile Detection | | | | | | | | ✓ | ✓ | ✓ | ✓ | | | | | ✓ | | | | ✓ | | | | | | |
| Sybil of Fake profile Detection | ✓ | | ✓ | ✓ | | | | ✓ | ✓ | | | | | | ✓ | | | | | | | | | | ✓ | ✓ |
| Detect information and location Leakage | | | | | ✓ | | | | | ✓ | ✓ | ✓ | | | | | | | | | | | | | | |
| Improving Privacy Setting Interfaces | ✓ | | | | ✓ | | | | ✓ | ✓ | ✓ | ✓ | ✓ | | ✓ | | | | | | | | | | | |
| NetNanny | | | | ✓ | | | | | ✓ | | | | ✓ | ✓ | ✓ | | | | | | | | | | | |
| Malware Detection | ✓ | | ✓ | | | ✓ | | | | | | | | | | | | | | ✓ | | | | ✓ | | |
| ZoneAlarm Privacy Scan | | | | | ✓ | | | | ✓ | ✓ | ✓ | ✓ | ✓ | | ✓ | | | | | | | | | | | |
| HRSP Url classification model | | ✓ | | | | | ✓ | | | | | | | | | | | | | | | | | | | |
| Defensio | ✓ | ✓ | ✓ | ✓ | | ✓ | ✓ | | | | | | | | | | | | | | | | | | | |
| Steganalysis | ✓ | | | | | | | | ✓ | | | | | | | | | | ✓ | | | | | | | |
| Privacy Scanner for FB | ✓ | | | | ✓ | | | | ✓ | ✓ | ✓ | ✓ | ✓ | | ✓ | | | | | | | | | | | |
| MyPermission | ✓ | | | | | | | | | ✓ | ✓ | | | | | | | | | | | | | | | |
| NoScript Security Suite | ✓ | ✓ | ✓ | | | | ✓ | | | | | | | | | | | | | | | | | | | |
| McAfeeSocialProtection | | | | | | | | | | ✓ | ✓ | | | | | | | | | | | | | | | |
| Norton Safe Web | ✓ | ✓ | ✓ | ✓ | | ✓ | | | | | | | | | | | | | | | | | | | | |
| FB phishing Protector | | ✓ | ✓ | | | | ✓ | | | | | | | | | | | | ✓ | | ✓ | | | | | |
| Minor Monitor | | | | ✓ | | | | | ✓ | | | | ✓ | ✓ | ✓ | | | | | | | | | | | |
| Watermarking | | | | | | | | ✓ | | | | | | | | | | ✓ | | | | | | | | ✓ |
| Report Users | ✓ | ✓ | ✓ | ✓ | ✓ | ✓ | | ✓ | ✓ | | | | ✓ | | ✓ | | | | | | | | | | | |
| MPAC Model | | | | | | | | | | ✓ | | | | | | | | | | | | | | | | |
| Internal protection Mechanism | ✓ | ✓ | ✓ | ✓ | | | | | ✓ | | | | ✓ | | ✓ | | | | | | | | | | | |
| Security and Privacy setting | ✓ | | ✓ | ✓ | | | | | ✓ | ✓ | ✓ | ✓ | ✓ | | ✓ | ✓ | | | | | ✓ | | | | ✓ | ✓ |
| Authentication Mechanism | ✓ | ✓ | | | | | | | ✓ | ✓ | | | ✓ | | | | | | | | | | | | | |
| AVG Privacy Fix | ✓ | | | | ✓ | | | | ✓ | ✓ | ✓ | ✓ | | | | | | | | | | | | | | |
| Co-ownership | | | | | | | | | | | | | | | | | ✓ | | | | | | | | ✓ | |
| Internet Security Solutions | ✓ | ✓ | ✓ | ✓ | | | ✓ | | | | | | | | | | | | | | | | | | | |
| Digital Oblivision | | ✓ | ✓ | | | ✓ | | | | | | | | | | ✓ | | | | ✓ | | | | | | |
| HRSP Content Model | | | | | | | | | | | | | | ✓ | | | | | | | | | | | | |
| Storage Encryption | | | | | | | | | | | | | | | | | | | | | | | ✓ | ✓ | | |
| Metadata removal and security | | | | | ✓ | | | | | | | ✓ | | | | | | ✓ | | ✓ | | | | | | |

### Internal protection mechanism

Some social networks provide a defense mechanism to protect their users from spammers, fake profiles, and scammers. For instance, Facebook provides an immune system known as FIS that keeps its users safe from malicious attacks.

### Security solutions

An increasing number of usages of OSN caused many security attacks as well. There are many solutions proposed to eliminate these threats. Some solutions can help the users of OSN that include the following:

*Watermarking.* Watermarking hides the text or image information to protect the original text or image. A single watermark isn't suitable for some applications. Instead, using multiple watermarks can enhance the robustness and increase the security of the image or any digital text. Discrete cosine transform (DCT), backward propagation of errors (BPNN), discrete wavelet transform (DWT), Elliptic curve cryptography (ECC), Singular value decomposition (SVD), and selective encryption methods are used to increase the robustness and security, but image quality is not degraded for several different signal processing attacks. After applying the particular encryption algorithm to the image, decrypting algorithms are applied to the watermarked image. These methods decompose the host image into third-level DWT (*Singh et al., 2018*).

*Co-ownership.* In the co-ownership model, several users used the privacy settings for multimedia content like pictures and videos. CooPed system is used for co-ownership that manages the access control for co-owned data. Its focus is on image-based objects, which consist of different parts and backgrounds, such as:

$$Object_j = \sum_i Object_j.Part_i + Object_j.Background.$$

The CooPed system is based on the SoNeUCONABC, a usage control model used to manage the access control. In that SoNeUCONABC access model, the user manages all objects he uploads and is the owner of these objects. By using CooPed system, a user can manage each $Object_j.Part_i$ which is related to him so that the user becomes the co-owner of the object. For all objects, $Object_j.Background$ is a fixed part that is managed by its owner and also related to him. The owner and co-owner have access policies, and if they grant access, then any other user can see the image or any content. In SoNeUCONABC, all objects $O_i$ is decomposed as $O_i^j . O_i$ is represented as a tree structure in which $O_i^j$ represents the leaf in that tree. An object decomposes into several objects (B, P1, P2, P3), in this, B represents the background (*González-Manzano et al., 2014*). There are some access control policies in which all users want that their established policies are enforced in which $O_i^j$ is used.

*Steganalysis.* On OSN, users can upload high-resolution pictures and videos, whereas OSN attackers can use this content as disguised objects and circulate malicious information. Steganalysis software is used to find that information within the bounds of that multimedia content. Many OSN users do not use such software tools as well as do not notify the

output because many steganalysis mechanisms are used to recognize the malicious pictures derived from supervised machine learning techniques. In these supervised machine learning techniques, a large number of datasets of pictures are collected to train a general model. By using that trained model, these pictures can be classified.

*Digital oblivion.*  Digital oblivion is a method in which an expiration time is applied on the OSN content like pictures or videos, so no one can access the data when the time is expired. The content or data is increasing in OSN by increasing the number of users, which needs expansion in the storing capacity along with its protection. The method of digital oblivion is used to protect the user's data. To provide digital oblivion, **X-pire!** is a tool to upload users' data like pictures, utilized with an expiration date on OSN like Flickr and Facebook. No one can access that data when the expiration date is reached, while users can extend the expiration date on their data.

*Storage encryption.*  Many OSNs store users' data in a third-party data center, while various OSNs do not have their data centers. These data centers store the users' data, can share it with any other organization without permission. The user's data on OSN have some sensitive information that is not publicly accessible, but if such information is exposed in public, the reputation of the user can be damaged that affecting him mentally as well as economically. The users' data should be stored in an encrypted form to protect, and no one can access it like an attacker or any other organization. Decentralized OSNs allow users to share their data on OSN without losing control over their data (*Chowdhury et al., 2015*). Many cryptographic techniques are used to protect the user's data on OSN.

*Metadata removal and security.*  Various approaches are used to remove the metadata and reduce the leakage of metadata privacy in OSN. One method is used for modifying the metadata in a file where the user can request to edit the metadata in a file with editing, and the user can produce an updating file. There is one technique for the security of multimedia metadata in which multimedia metadata is encrypted and stored in the multimedia file.

*Intrusion detection.*  In OSN, unauthorized users can be detected by an intrusion detection system. This detection system is used to analyze the actions of users and the system; any disallowed action can determine the unauthorized user. It also detects the user's behavior, whether they act normal or not. Behavioral intrusion detection is used to detect the behavior of the system. The host and network generate the system behavior, and the user behavior is related to the interaction between the user and the system (*Peng, Choo & Ashman, 2016*). Principal component analysis (PCA) analyzes user behavior and detects anomalies (*Viswanath et al., 2014*).

## Commercial solutions

Due to several threats to OSN, many commercial companies make software to remove threats to protect the OSN users. Users must pay a fee for using these software solutions (*Fire, Goldschmidt & Elovici, 2014*). All commercial solutions for software are displayed in Fig. 5. A description of commercial solutions is presented in Table 8.

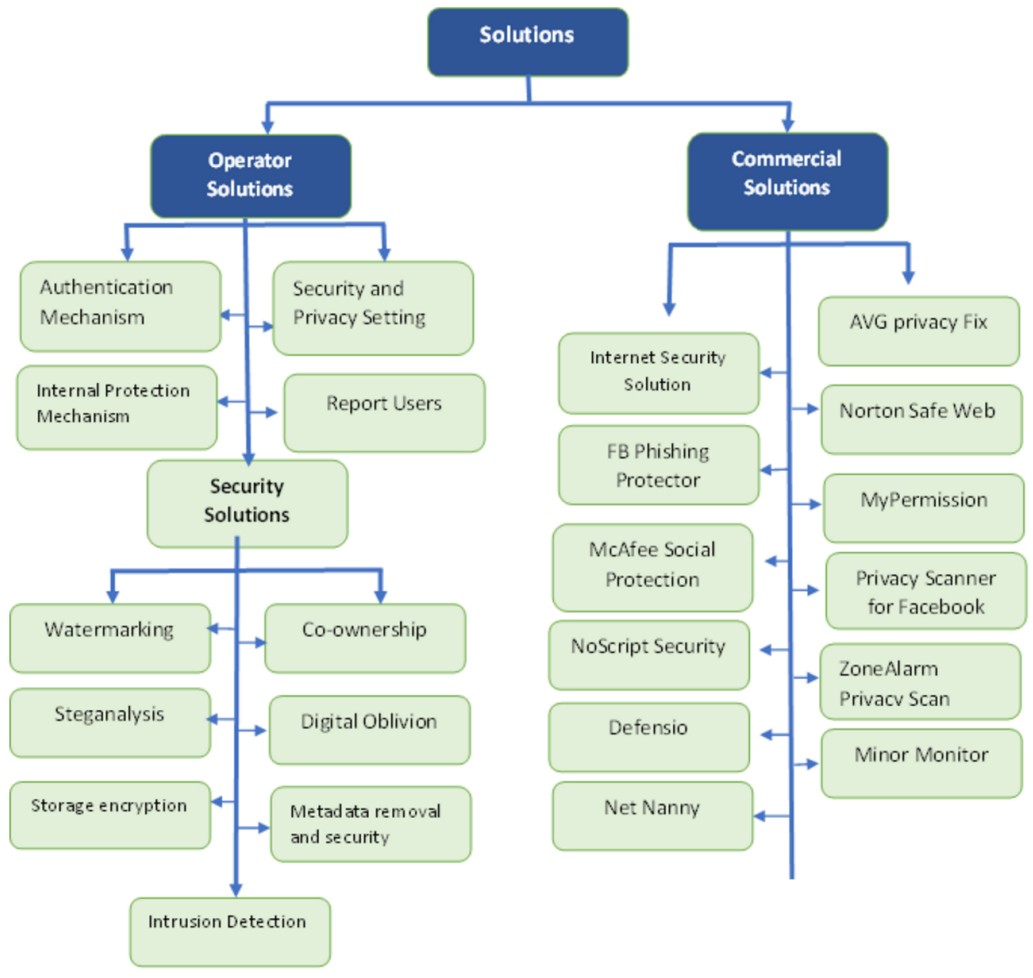

**Figure 5** Solution of threats 1.

### Internet security solution

Security companies have software solutions to protect against attacks. These companies like McAfee, Panda, Symantec, *etc.*, offer antivirus and firewalls that protect the user's system.

### AVG privacy fix

AVG privacy Fix software is used on mobile devices as an application or on web browsers for Facebook, Google, and LinkedIn to protect the user's privacy.

### FB phishing protector

FB Phishing Protector is a tool of Firefox that everyone can freely use to protect them from attacks. This tool prevents users from phishing scams on Facebook. The user needs to install an add-on for use on Facebook. If any site is trying to obtain the user's login and password or running a malicious browsing script at that time, FB Phishing Protector instantly informs the user about malicious activity (*Umar, 2011*).

### Norton safe web

Users used Norton Safe Web to identify websites that used malicious content or links. This service is free for users and informs them which site is safe to visit.

### McAfee social protection

McAfee announces McAfee Social Protection for Facebook, allowing users to share photos only with selected friends.

### MyPermission

MyPermission is a privacy cleaner that scans the user's applications and data to discover threats in it.

### NoScript security suite

No Script Security Suite has a blocking approach that blocks the malicious scripts running on the browsers. This software is free for Google Chrome, Sea Monkey, and Mozilla Firefox. The software allows enabling JavaScript on browsers.

### Privacy scanner for facebook

Facebook used privacy scanners to protect its users. ReclaimPrivacy and Trend Micro Smart Protection scanners are used for Facebook. ReclaimPrivacy tool checks the privacy setting for any privacy issue, and if there is an issue, it offers the solution. Trend Micro Scanners are used to block malicious activity and links on Facebook (*JubileeX, 2020*).

### Defensio

Defensio is a web security service that removes malicious content, spam, and comments from the web. This service prevents many social threats to OSN users, like removing malicious content and URL-based threats (*Crunchbase Defensio Organization, 2007*).

### ZoneAlarm privacy scan

ZoneAlarm is an anti-phishing privacy scan that is used to detect phishing attacks and the security of its users. It is a chrome extension that prevents users from entering any sensitive information.

### Net Nanny

Net Nanny is software that controls the content on websites and social accounts. It provides age-based filtering and blocks inappropriate content. If a kid tries to open a secured application, Net Nanny prevents launching that application.

### Minor Monitor

Minor Monitor is a free service that parents use to track and monitor their children's Facebook activities to keep them safe.

## Academic solutions

Several solutions for malicious activities are proposed for OSN users that prevent them from attackers. The main diagram of the academic solutions is shown in Fig. 6. These solutions include the following (*Fire, Goldschmidt & Elovici, 2014*):

 

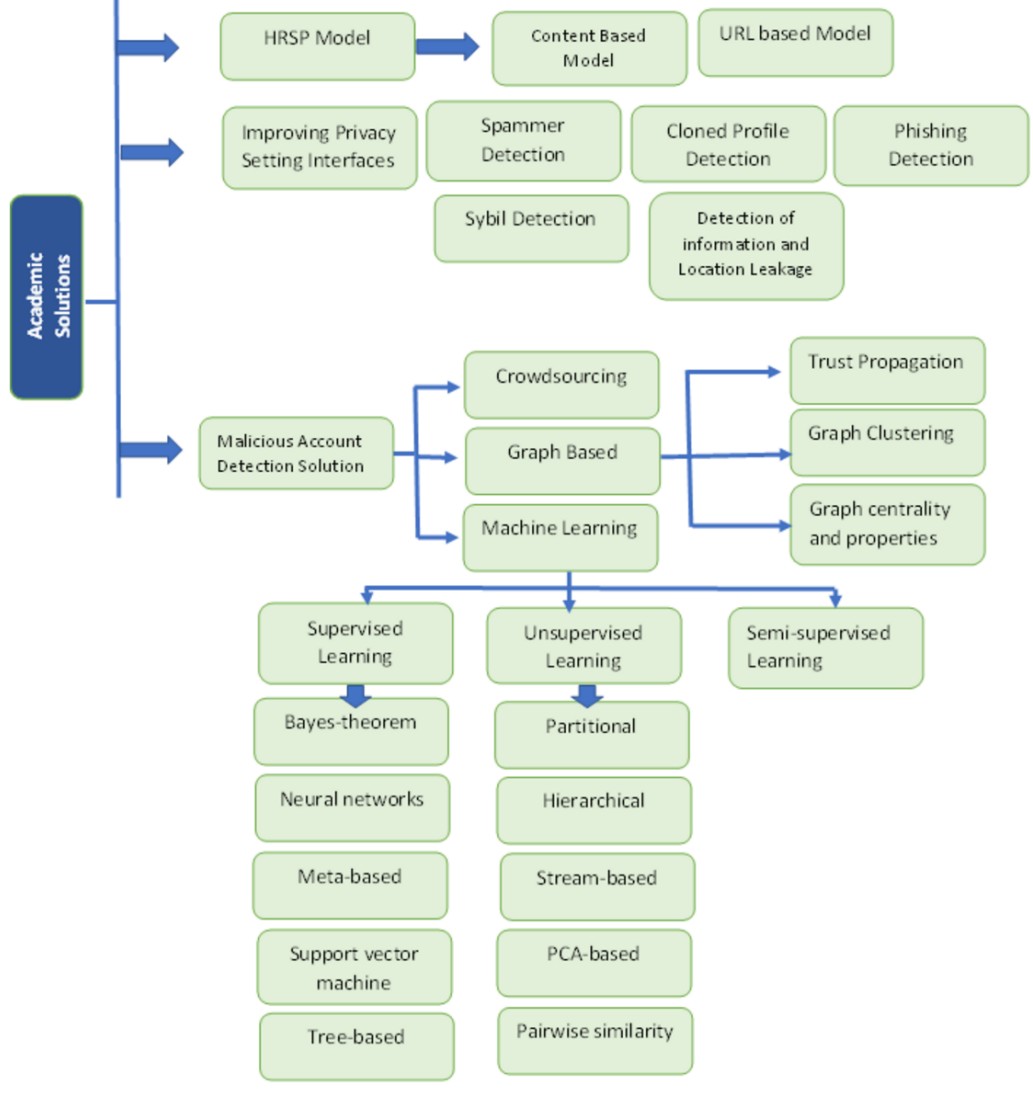

**Figure 6  Solution of threats 2.**

### Improving privacy setting interfaces

Solutions proposed for OSN users have increased recently, whereas many solutions and apps are available to protect users' privacy. For example, on Facebook, users need to know who sees their photos, posts, contents, or friend list after some privacy settings. Facebook offers the function in which one user sees the interface of their account that can see by an unknown user.

### Phishing detection

Companies developed software, and researchers proposed solutions for phishing attacks. Countermeasures for phishing attacks include machine learning, text mining, human users, and profile matching (*Aleroud & Zhou, 2017*). A real-time anti-phishing system that uses seven algorithms such as K-nearest neighbor (kNN) ($n = 3$), random forest, naïve

Bayes, Adaboost, decision tree, sequential minimal optimization (SMO), and K-star and compared the performance of these algorithms (*Sahingoz et al., 2019*).

### Spammer detection

There are some spammer detection techniques proposed for users to detect spam. A community-based feature for identifying spammers involves two types of detection: Node Level Community detection and Feature Extraction (*Bhat & Abulaish, 2013*). Moreover, another feature selection approach to detect spam on Facebook uses a Particle Swarm Optimization algorithm (*Sohrabi & Karimi, 2018*).

### Cloned profile detection

There are two approaches based on attribute similarity and similarity of friend networks to detect cloned identities (*Jin, Takabi & Joshi, 2011*). Attribute similarity is derived from the two profile attributes' similarity and their value, whereas the friend similarity network measure calculates the similarity between two identities' friend networks. Based on these, two more similarity measures, basic profile similarity, and multiple fake identities profile similarity are available. In basic profile similarity, it is assumed that attackers do not make fake identities while friends involving victims have authentic identities. The measure model has been examined in two steps, including the formative and reflective constructs. The questionnaire method is used for identity theft, and the partial least squares method is used for data analysis. Technological and conventional copying is specified as reflective constructs in which hypotheses and questionnaires decrease the effect of identity theft (*Lai, Li & Hsieh, 2012*).

### Sybil detection

Attackers create patterns that the machine cannot understand, and recently several machine learning algorithms have been used to detect and solve fake profile problems. These machine learning algorithms include random forests, naive Bayes, Adaboost, and support vector machine (SVM) (*Ramalingam & Chinnaiah, 2018*). A new random walk-based method for Sybil attacks is called SybilWalk (*Jia, Wang & Gong, 2017*). Using the method including labeled benign nodes and labeled Sybil, there is no random walk-based Sybil detection method, so label augmented social network is used for these two labels. Users of Twitter and Facebook are treated as Labeled benign, and the users or attacker spreading the malicious contents is labeled Sybil. On the dataset, Twitter Sybilwalk attain a positive rate of 1.3% and a negative rate of 17.3%.

### Detection of information and location leakage

Twitter offers the guardian angel service that tells the user if any privacy violation occurs. Location leakages in mobiles are common. To overcome this threat, a mechanism proposed for location sharing privacy is BMobishare (*Shen et al., 2016*), which uses two cryptography schemes: public-key encryption and symmetric key encryption and hash function.

### Malicious account detection solutions

There are three methods to detect suspicious accounts on OSN: crowdsourcing, graph-based, and machine learning given below (*Adewole et al., 2017*).

*Crowdsourcing.* The crowdsourcing method is used to detect malicious accounts, which involves human detection to check the profile of users to detect any malicious activity. Several numbers of tasks are distributed among internet users who detect suspicious patterns.

*Graph-based.* The graph identifies the malicious behavior and is represented as G = (V, E), whereas V is the set of vertices, and E is for edges. In OSN, this graph is called a social graph (*Adewole et al., 2017*). In the G graph, edges and nodes can vary according to the problem in which edge represents the friendship invitation. A social graph is unipartite means it has one type of node; if the social graph is bipartite or tripartite, it has multiple types of nodes. The extension of the social graph increases as the number of edges and nodes increases. Following are the three methods used for graph-based algorithms.

*Trust propagation.* There are two types of trust relationships in OSN: weak and strong. The graphs of OSN that have strong trust hold the fast-mixing property, whereas honest and Sybil are the two regions of the graph. OSN, which has a strong relationship, has several attack edges among honest and Sybil regions, while having a weak relationship does not hold the property of fast- mixing. The trust propagation method evaluates a degree-normalized landing probability and OSN assigned to every node. This probability correlates with a modified random walk that lands on each node and starts from the non-Sybilnode. Non-Sybilnode spreads its value to the neighboring nodes. A stochastic process finished in the early phase is known as a short walk. A random walk that runs for a prolonged period will create uniform trust rank values in OSN for all nodes. This value is called the convergence value of a random walk.

*Graph clustering.* In graph clustering, similar nodes can make the group of a set if their distance is specific to each other. The nodes that make a group are called clusters, and it is also called communities. The algorithm used for graph clustering is the Markov Cluster (MCL). On the graph, MCL constantly clusters nodes and terminates when the stable matrix is obtained. The cluster obtained in the result can be used to analyze the malicious account.

*Graph Metric and properties.* The importance of nodes is measured in graph centrality based on their position in the social network. On the social graph, the node is located on a social path among other nodes determined by centrality betweenness. Metric shows the percentage of all shortest paths in the network, whereas PageRank, closeness, and eigenvector are the centrality metrics implemented in the graph theory. Some interesting properties of a social graph include scale-free topological structure, power-law distribution, and small-world, which includes graph centrality to help distinguish harmful accounts.

*Machine learning.* Machine learning (ML) is used to identify malicious accounts using Supervised, Unsupervised, and Semi-supervised learning methods, which further have sub-sections.

*Supervised learning.* In ML, Supervised learning is a function in which input is mapped to the output as an input–output pair. The input may be a vector, and the output may be a supervisory signal. A common example of supervised learning is linear regression in regression problems. In supervised machine learning, training data is analyzed to produce a classification model for predicting unseen data. The classification model is used for detecting malicious accounts at the end of training. The following categories are involved in it:

i. ***Bayes theorem:*** In Bayes' theorem, the probability of a hypothesis is described under given conditions. This theorem shows how much proof influences the probability that a given hypothesis is correct. In various domains, the Bayes theorem has an application that starts from topic modeling to spam filtering in the social network. On top of the Bayes Theorem, Bayesian Network and naiveBayes are built to detect malicious accounts and URLs in social networks and perform well. Hypothesis H is given, and evidence E, Bayes' theorem declares that the relationship between the probability of the hypothesis before obtaining the P(H) evidence and the probability of the hypothesis after obtaining the evidence P(H\E) is

$$P(H|E) = P(E|H).P(H) \; P(E)$$

ii. ***Meta based:*** In supervised learning, meta-based classifier is used to enhance the generalized ability of learned models. Based on the nature of the data set, a meta-based classifier is used to predict the classifier that is good for a given task. The meta-based classifier uses other classification algorithms and does not use its own to perform the task. Hence, this classifier is used to help the users for choosing an algorithm suitable for their given problem.

iii. ***Support vector machine:*** A support vector machine (SVM) is used to detect malicious accounts, conserving high-performance accuracy and deducting the errors. This machine is used to analyze the data and also detect patterns by using label samples. At AT&T Bell Laboratories, the support vector machine evolved, which can be used for classification and regression problems. By explaining a separating plane, a support vector machine is used to isolate the boundary among different classes in the dataset called a hyperplane. SVM uses kernel functions of nonlinearly separable problems to obtain optimal separating hyperplane.

iv. ***Neural networks:*** In speech processing, image processing, pattern recognition, and disease diagnoses, in all these application domains, the neural network has been used. Because the neural network has a high computational requirement, social networks found little application in detecting malicious accounts. The neural network includes a multilayer perceptron (MLP). MLP contains activation units, mentioned as artificial neurons and weights, and multilayer perceptron is the class of free-forward artificial neural networks. By involving multiple layers, multilayer perceptron improved the standard linear perceptron like input, hidden, and output layers that are used to solve linear and nonlinear classification problems. The algorithm is used to map the input data to the proper output.

v. ***Tree-based:*** An algorithm uses the ability of a decision tree in which the classifier is trained with the structure of a tree. The test of an attribute value shows the node

on the tree, and the test result is represented by the branch. Random forest and J48 (C4.5) are decision tree algorithms identifying phishing attacks and spam on online social networks. C4.5 is a decision tree algorithm developed by Ross Quinlan. C4.5 is an extension of the earlier Quinlan ID3 algorithm. C4.5-generated decision trees can be used for classification. On C4.5, J48 is based, and this algorithm is the extension of Iterative Dichotomiser 3 (ID3). ID3 is the algorithm used to make a decision tree from the dataset and is typically used in natural language processing domains and machine learning. To select the most suitable attribute at each node of the tree, C4.5 uses the information gain. That attribute shows the best candidate used to decide on the splitting of the tree. By creating diverse decision trees applying random feature selection and bagging approach at training time, Random Forest produces an ensemble of the classifier. The decision tree has two types of nodes, the first is the leaf node labeled as a class, and the second is the interior node related to the feature.

*Unsupervised learning.* Unsupervised learning is an artificial intelligence algorithm that uses information which is not labeled and classified. The algorithm performs on that information without any guidance. The unsupervised learning groups the data based on similarity. The unsupervised learning methods include:

i. *Hierarchical:* Using a tree structure, Hierarchical clustering (HC) groups the data over a variety of scales. The tree is a multilevel hierarchy to obtain clusters at the next level; clusters present at one level are merged or split. Hierarchical clustering is either bottom-up called agglomerative or top-down called divisive. By using the bottom-up approach, agglomerative clustering builds hierarchy, presuming that each instance originally forms its cluster. The algorithm then continually merges the pairs of clusters while one moves up the tree. The top-down divisive approach worked contrarily and presumed that each instance remains in the beginning in one cluster. The algorithm constantly sorts the cluster while it traverses down the tree.

ii. *Partitional:* A cluster is defined when each set of instances are split in such a way that there remains no overlapping. K-means algorithm is one example of a partitional clustering method with various application areas. K-means, a heuristic clustering algorithm, clusters the dataset into user-defined K clusters by reducing the amount of squared distance in each cluster. Using the K-means algorithms, it is required to compute the distance between a point to its centroid.

iii. **PCA-based:** For spotting patterns in high dimensional data, principal component analysis is used as a statistical tool. The principal component analysis (PCA) is used for detecting the variations in a dataset. It is a principled candidate for detecting destructive behavior in online social networks.

iv. **Stream-based:** The stream-based approach is stimulated by the issuance of the stream clustering algorithm used to separate malicious accounts from legitimate accounts. Stream and StreamKM++ are the two stream-based clustering algorithms that are used to identify malicious accounts on Twitter. Stream, a clustering algorithm, expands the traditional batch learning that is the DBSCAN algorithm by explaining the core-micro-objects instead of the core objects concept used in DBSCAN. StreamKM++ algorithm

is developed because the K-means algorithm needs a predefined number of clusters and a random initial centroid selection.

v.   **Pairwise similarity:** The pairwise similarity detect malicious activities by comparing two accounts. This method is used to detect anomalies in the online social network. If the user's account is compromised, study the legitimate user's behavior history, which is used for a particular period. Investigate that the clickstream activities utilized the user's extroversive and introversive social behavioral pattern to form the effective behavioral model. Euclidean distance is applied to determine the difference between the two profiles. Assume that both profiles are P and Q, consisting of extroversive and introversive feature vectors. Suppose A =(a1, a2, a3,…an) and B = (b1, b2, b3,…bn) indicate the feature vector for P and Q. Euclidean distance among A and B is evaluated as shown in Eq. (a) and Eq. (b) that indicates the computation of Euclidean norm among profiles P and Q on the Euclidean distance for all feature vector. If the value of the distance is higher, the more significant two profiles differ. 'm' in Eq. (b) indicates the number of the feature vector. There are eight extroversive and introversive behavior considered.

$$E(A, B) = \sqrt{\sum_{k=1}^{n}(a_k - b_k)^2} \qquad (a)$$

$$Dist(P, Q) = \sqrt{\sum_{f=1}^{m}(E_t)^2} \qquad (b)$$

*Semi-supervised learning.*   During training, Semi-supervised learning amalgamates the smaller amount of labeled data with the larger amount of labeled data. It is between unsupervised learning with no labeled training data and supervised learning with labeled training data. Many semi-supervised algorithms, such as self-training, expectation–maximization, co-training, and transudative support vector machines (TSVM), detect phishing attacks (*Li et al., 2013*). Image and document object model (DOM) features are used to train the TSVM algorithm, whereas an evolutionary algorithm (QEA) deals with the local convergence problem of TSVM. SVM of TSVM outperformed its counterpart, which has an accuracy of 95.5. By inspecting both labeled and unlabeled data, previous experiments show that TSVM can upgrade in execution over its SVM counterpart.

### Hybrid real-time social networks protector (HRSP) system model

HRSP system model provides simultaneous user-level protection against OSN attacks and threats. To achieve this goal, HRSP provides a solution for two groups of OSN threats uniform resource locator (URL) based model and the content-based model (*Yassein, Aljawarneh & Wahsheh, 2019*). User Level Security Protocol (USP) and HRSP system proposed a user-level security protocol for OSN. By using cryptographic techniques, USP

is used to add user-level security features. USP has a common structure for protecting users from the account and content-based threats. Due to the weakness of the current security service in OSN, the USP protocol is proposed, which uses cryptographic algorithms with some security services like Digital Signature (DS) and Data Encryption (DE).

*HRSP URL classification model.* Many URL-based threats like phishing and XSS cause problems in OSN. The user-level security protocol is the solution for these threats. If USP fails to protect the user, they need to check the user's content like shared posts and check if there are any URLs. There are many solutions, but they are not fulfilling all aspects like Blacklist-based solutions can only detect the malicious URLs, and feature-based classification causes network delay. URL lexical features are analyzed in this model, which is the best available feature. This model does not check the contents of the web page, DNS, or WHOIS to reduce the network delay. An attacker can use the URL shortening services to redirect users to malicious web pages by bypassing the blacklists and hiding the URL. The purpose of hiding is that the user cannot find the redirected URL.

*HRSP content classification model.* The content classification model is used to analyze the user's content, like posts and comments, to eliminate the threats. The content classification model is not easy to develop because plenty of content needs to be analyzed in many languages. A core hybrid system for Twitter OSN is based on users' feedback with expandable solutions. For both URL and content classification models, the HRSP browser's extension should have the facility to process the content directly without the need to copy and paste the content. But users have the opportunity for feedback on the content model, and 13 features are used to construct the HRSP content classification model.

## FUTURE DIRECTIONS

Security is a major concern in the OSN sector and involves storing user personal information and performance in various accounts, for which investigators offer a variety of solutions to protect personal information from attacks and threats. Every day, attackers create new tactics and gateways into the new user's account to collect and pick out information about the user. Currently, a single fake profile attack or clone profile attack spreads rapidly on any social network to steal user privacy. Investigators use irregularity detection algorithms to track compromised accounts on social networks such as Facebook and Twitter. Researchers offer several solutions for detecting fake profiles, but existing solutions cannot prevent account creation or incorrect profiles. Similarly, some researchers use machine learning methods to analyze and detect various malicious activities and draw conclusions to analyze the activity of a profile. The data leakage algorithm is also used to monitor the performance of profiles for analyzing posts and ads on social media.

A way to improve the privacy and security of profiles on OSN is to use an integrated model of an existing mechanism to protect profiles from malware, identity theft, malicious account recovery, and other attacks. For this purpose, a generic framework is needed to prevent the creation of fake profiles on any platform. Whenever a friend requests a friend account, the framework should pass through the account correctly and sends a message

to the recipient's account whether the profile is fake or legitimate. Given these concerns, another standard framework that will protect user privacy on any platform by incorporating specific encryption should be developed. Overall, performance measurement is a major concern in the current situation where researchers are integrating their frameworks and models into different OSNs. The above concerns require a performance inspector to determine the performance and implementation of various frameworks and security.

## CONCLUSION

This study discusses about the behavior of users on online social networks and discusses available services that can help secure usage of social media platforms. It also focuses on different threats posed by sharing multimedia content across social networks and provides mitigating strategies. Furthermore, it discusses users' behavior on different social networks. It is anticipated that there is a need to investigate users' behavior on OSNs to create exciting research stories and other interesting solutions in this area. It is expected that this research will help guiding social media users to improve their usage and help the social media platforms provide better user satisfaction.

## ACKNOWLEDGEMENTS

The authors acknowledge the efforts of Zunaira Nawaz as she helped to improve the language of the article along with the suggestions to improve the related work section. Moreover, she helped in gathering relevant research articles.

### Funding
The authors received no funding for this work.

### Competing Interests
Adnan Abid is an Academic Editor for PeerJ Computer Science.

### Author Contributions
- Naeem A. Nawaz analyzed the data, authored or reviewed drafts of the article, and approved the final draft.
- Kashif Ishaq conceived and designed the experiments, analyzed the data, prepared figures and/or tables, authored or reviewed drafts of the article, and approved the final draft.
- Uzma Farooq performed the experiments, authored or reviewed drafts of the article, and approved the final draft.
- Amna Khalil conceived and designed the experiments, performed the experiments, performed the computation work, prepared figures and/or tables, and approved the final draft.
- Saim Rasheed performed the experiments, performed the computation work, prepared figures and/or tables, authored or reviewed drafts of the article, and approved the final draft.

- Adnan Abid conceived and designed the experiments, authored or reviewed drafts of the article, and approved the final draft.
- Fadhilah Rosdi analyzed the data, authored or reviewed drafts of the article, and approved the final draft.

## Data Availability

This is a literature review.

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
