# Peer review of "A comprehensive review of security threats and solutions for the online social networks industry"

_PeerJ Computer Science, doi:10.7717/peerj-cs.1143_

## Round 0.1 · original submission · Major Revisions

The referral process is now complete. While finding your paper interesting and worthy of publication, the referees and I feel that more work could be done before the paper is published. My decision is therefore to provisionally accept your paper subject to major revisions.

The referees' comments are given on the website. Please make sure that you have addressed all the comments.

Since this is a survey paper, more discussion is needed. Moreover, a systematic literature review approach should be used.

Reviewer 1 ·

Basic reporting

This review if within the scope of the journal, and can be useful for the related community.

Experimental design

The protocol of the research was not well described. Without it, it is hard to assess the study’s validity and better understand its scope. The systematic review research protocol is much more than “research questions” and "search engine." Please consult the literature regarding systematic mapping and describe this protocol at the beginning of section 3.

It is also unclear where the security threats and techniques/solutions were searched. The search string and the chosen databases show that only scientific papers were analyzed. However, there are a considerable number of commercial tools, that usually do not appear in this type of document.

Validity of the findings

The validity of the finding is severally compromised due to the lack of the systematic mapping protocol description. The authors should better define this protocol, describing for example inclusion criteria. After that, it will be possible to define if the research can be considered an interesting contribution to the area.

Additional comments

The paper is well written, congratulations to the authors. However, please avoid strong adjectives, such as “immense data usage.” They are hard to be defined and verified.

Reviewer 2 ·

Basic reporting

- This is a well structured manuscript.
- The language usage and grammer is acceptable.
- According to the authors’ goals, the manuscript has supported arguments.
- At the line 118: Instead of writing “It says” it is better to use “He says” or “It is stated that”.
- Throughout the manuscript, some well known abbreviations have been used such as kNN, SMO, SVM, etc. Even though they are well known, again the open format of those abbreviations should be given when they are firstly mentioned in the article. For example;
• At the line 540: kNN has been used,
• At the line 541: SMO has been used,
• At the line 562: SVM has been used.
- Future directions are also mentioned in the manuscript.

Experimental design

The study design of the manuscript is acceptable.

Validity of the findings

Acceptable.

Additional comments

-

---

## Round 0.2 · accepted · Accept

Since the reviewers' comments have been addressed in detail, it's my pleasure to inform you that your manuscript has been accepted.